# Health Literacy among People in Cardiac Rehabilitation: Associations with Participation and Health-Related Quality of Life in the Heart Skills Study in Denmark

**DOI:** 10.3390/ijerph17020443

**Published:** 2020-01-09

**Authors:** Anna Aaby, Karina Friis, Bo Christensen, Helle Terkildsen Maindal

**Affiliations:** 1Department of Public Health, Aarhus University, Bartholins Allé 2, 8000 Aarhus C, Denmark; bc@ph.au.dk (B.C.); htm@ph.au.dk (H.T.M.); 2DEFACTUM, Central Denmark Region, Olof Palmes Allé 15, 8200 Aarhus N, Denmark; Karina.friis@stab.rm.dk; 3Steno Diabetes Center Copenhagen, Niels Steensens vej 2, 2820 Gentofte, Denmark

**Keywords:** health literacy, cardiac rehabilitation, cardiac disease, rehabilitation participation, health-related quality of life, physical health status, mental health status, health equity

## Abstract

Health literacy (HL) is a dynamic determinant of health and a promising target of health equity interventions in noncommunicable disease prevention. Among people referred to a cardiac rehabilitation program, we examined the associations between (1) HL and participation in cardiac rehabilitation and (2) HL and health-related quality of life (HRQoL). Using a cross-sectional design, we invited 193 people referred to cardiac rehabilitation in Randers Municipal Rehabilitation Unit, Denmark, to respond to a questionnaire in 2017. Of these, 150 people responded (77.7%). HL was measured using the nine scales of the Health Literacy Questionnaire (HLQ), while HRQoL was measured using the Short Form Health Survey 12 (version 2) (SF-12). The mean age of respondents was 67.0 years; 71.3% of the sample were men. Nonrespondents had significantly lower educational attainment and more often lived alone than respondents. Using multiple regression analyses, we found no significant associations between HL and participation in cardiac rehabilitation. There were significant positive associations between several aspects of HL and physical and mental HRQoL. HL could be a factor of interest in initiatives aimed at improving participation and outcomes of cardiac rehabilitation.

## 1. Introduction 

Cardiac rehabilitation has proven effective in improving quality of life and preventing adverse events after cardiac disease onset [1]. However, participation rates are low in many European countries [2]. Studies have shown that participation and adherence is largely dependent on demographic and social circumstances such as age, gender, cohabitation status, educational attainment, work situation, and income, as well as practicalities such as distance to rehabilitation facilities [3]. From the perspective of a rehabilitation service unit, few of these factors are easily modified or targeted. Thus, identification of more dynamic intermediary determinants is crucial to encourage increased participation in cardiac rehabilitation and respond to the needs of those who do not attend [2,4]. 

Health literacy is defined as “the combination of personal competencies and situational resources needed for people to access, understand, appraise and use information and services to make decisions about health” [5]. Thus, health literacy encompasses a broad spectrum of cognitive and social skills enabling an individual’s navigation, motivation, and action competence in relation to health promotion, prevention, and care [6]. Moreover, health literacy is a dynamic quality formed and executed through people’s interactions with the surrounding environment, including healthcare organizations and preventive services [7]. 

Health literacy has been identified as a mediator between social health determinants and health outcomes [8,9,10,11]. In relation to cardiac conditions, low health literacy has been associated with higher prevalence [12], poorer cardio-protective health behaviors [13,14], less disease knowledge and poorer self-management [14,15], higher hospital readmission rates [16,17,18], and higher mortality [17,18]. In addition, low health literacy has been associated with low health-related quality of life (HRQoL) [13,16,19,20,21], particularly in relation to the physical components of HLQoL [19,20]. Improved HRQoL is an important and well-supported outcome of cardiac rehabilitation programs [1]. The effect is consistent across many types of rehabilitation programs [1], suggesting that other factors than specific program elements could play a role. However, to the best of our knowledge, no studies have examined the association between health literacy and HRQoL in the context of cardiac rehabilitation. 

Few studies have evaluated the impact of health literacy in a cardiac rehabilitation context. Dankner et al. found that lack of knowledge regarding cardiac rehabilitation was the most common reason for nonparticipation in cardiac rehabilitation [22]. Mattson et al. found that health literacy was associated with increased knowledge gain from cardiac patient education during a rehabilitation program [23]. A few intervention studies have also successfully addressed health literacy in their intervention design or measured it as an outcome [24,25]. In all these studies, as in most studies on health literacy in people with cardiac conditions, individual health literacy was measured using a single value or category [26,27,28,29]. These tools do not account for the diversity of health literacy assets and challenges within populations and thus suffer from an inability to be easily translated into appropriate responses at the individual and organizational level.

Among people referred to a municipal cardiac rehabilitation program, and using a broad measure of health literacy, the aim of this study was to
Examine the associations between nine aspects of health literacy and participation in cardiac rehabilitation.Examine the associations between nine aspects of health literacy and physical and mental HRQoL.

A subsidiary aim was to examine the association between socio-demographic variables (age, gender, cohabitation status, and educational attainment) and participation in rehabilitation as well as HRQoL and compare these results with the analyses on health literacy.

## 2. Methods

*Design and setting.* This cross-sectional study was part of the Heart Skills study aimed at developing a co-designed intervention responding to the health literacy needs of people referred to Randers Municipal Rehabilitation Unit, Denmark. We collected survey data from 193 people referred to the cardiac rehabilitation program in the unit. Randers Municipal Rehabilitation Unit offers free-of-charge nationally recommended cardiac rehabilitation programs [30] to citizens with a cardiac condition or severely adverse cardiac risk factors in Randers Municipality. Most of these are referred to a start-up session in the unit as part of their discharge from hospitals or later in their disease course from their general practitioner. They are then contacted by the unit, and if accepted, a session is scheduled. Nonparticipants may either decline the start-up session or chose not to attend further services after the information provided at the start-up session.

The municipality has approximately 98,000 inhabitants. The program includes a twelve-week biweekly exercise training program and eight patient education sessions, as well as optional elements including smoking cessation support and dietary, sexual, or energy management counselling. 

*Data collection.* All people referred to the cardiac rehabilitation program in Randers Municipality between 7 March 2017 and 31 December 2017 were eligible for the study. Verbal consent to participate in the survey was obtained shortly after referral as part of a telephone invitation to the cardiac rehabilitation program. The survey was then distributed to consenting individuals according to the participant’s favored mode of communication: by e-mail, by postal letter, by telephone interview, or on a paper questionnaire or tablet at the Rehabilitation Unit with support from a healthcare provider. After predetermined time intervals depending on the distribution method, nonresponders were reminded to complete the survey. After another predetermined time interval, remaining nonresponders were contacted by phone to confirm their consent and preferred survey format. Figure 1 shows the selection procedure.

The survey included questions on health literacy, HRQoL, health behavior and the disease course leading up to the referral. Prior to data collection, two people referred to the rehabilitation program and two people, who had completed the rehabilitation program tested the questionnaire in a small pilot. The test led to only minor linguistic changes. 

*Health literacy measure.* Health literacy was measured using the comprehensively validated 44-item Health Literacy Questionnaire (HLQ) [31,32,33]. The HLQ consists of nine scales that independently measure different aspects of health literacy (Table 1). The HLQ was translated into Danish following standardized forward and backward procedures. The Danish language validation study showed a robust nine-dimension confirmatory factor model [34]. 

In scales 1–5, respondents use a four-point scale: “Strongly disagree”, “disagree”, “agree”, and “strongly agree”. In scales 6–9, respondents use a five-point scale: “Cannot do”, “very difficult”, “quite difficult”, “quite easy”, and “very easy”. Mean scales scores are calculated and reported with standard deviations. If responses to more than two items in a scale were missing for a particular respondent, the scale score for that respondent was reported missing.

Over the past decades, the concept of health literacy has developed from a narrow term closely related to literacy and numeracy into a multifaceted concept developed and understood in relation to an individual’s context and the surrounding health system and community [5]. The HLQ measures health literacy from this broad understanding. Several attempts to further subcategorize aspects of health literacy are available in the literature [31,35,36]. Inspired by these studies and to support the interpretation of our results, we have attempted to categorize the nine HLQ scales into three categories: cognitive capacities, executive capacities, and social capacities (Table 1). 

*Health and rehabilitation measures.* Participation in rehabilitation and HRQoL were used as dependent variables. Respondents reported on their current or intended participation in the rehabilitation program. Only current participation was labelled as “participation in rehabilitation” in the analyses. 

HRQoL was measured using the validated Short Form Health Survey 12 (version 2) (four-week recall) (SF-12), which examines self-experienced health within eight domains of physical (four domains) and mental (four domains) health. Norm-based physical (PCS) and mental (MCS) component summary scores were calculated using the standardized procedures based on the general U.S. population [37]. 

Data on comorbidity was used as a covariate in the adjusted analyses. This was obtained from the Danish National Patient Registry (LPR) [38] and included all International Classification of Diseases 10th Revision (ICD10) codes obtained during hospitalizations or outpatient visits between 1994 and 2017. Data was reported as a weighted Charlson index score, which is based on 17 diagnostic categories [39,40]. Although the index contains several cardiovascular conditions, we included all categories to allow for comorbidity based on multiple cardiovascular diagnoses. We included data from the year of referral (2017), as the hospitalization most respondents had experienced in relation to their cardiac event would increase the likelihood of any relevant diagnoses being recorded for that individual. For analyses, data were dichotomized as a weighted Charlson index score of ≤1 or >1.

*Socio-demographic measures.* Age, gender, cohabitation, and educational attainment were used as independent variables *and* covariates in the adjusted analyses. Country of origin was only used for adjusted analyses, as the vast majority of survey participants were from Denmark. All data were obtained from registers at the central authority on Danish statistics (Statistics Denmark). Cohabitation, educational attainment, and country of origin were dichotomized. Any type of habitation that encompassed adults living together was termed cohabitation. In the case of educational attainment, we dichotomized 11 years of schooling or below versus above 11 years of schooling. Those individuals who were born or registered outside Denmark, or whose parents were born or registered outside Denmark, were classified as having a non-Danish country of origin. If one parent was Danish, the country of origin was based on the mother’s birthplace or registration.

*Statistical methods.* Descriptive information was reported using appropriate summary measures on the total population as well as to compare responders and nonresponders. 

Multiple logistic regression was used to analyze associations between the independent variables (age, gender, cohabitation status, educational attainment, and the nine HLQ scales) and cardiac rehabilitation participation. We used nonparticipation as the reference category, and reported our results as odds ratios (95% CI). Multiple linear regression was used to analyze the associations between the same set of independent variables and HRQoL for physical and mental component summary scores separately. The results were reported as β coefficients (95% CI). All regression analyses were adjusted for age, gender, country of origin, cohabitation status, educational attainment, and comorbidity, excluding the independent variable under examination in relevant cases. 

The level of significance was set at *p* < 0.05. All statistical analyses were performed using STATA version 15.1 (Metrika Consulting, Stockholm, Sweden).

*Ethics and approvals.* According to Danish law, no specific ethics evaluation was required for this survey study. The study was approved by the Danish Data Protection Agency (2015-57-0002 (62908, 141)). The study was performed in accordance with General Data Protection Regulation (GDPR) and the Helsinki Declaration. Information about the study aim, voluntary participation, and confidentiality was provided with the questionnaire. All participants gave their verbal consent to take part before the questionnaire was distributed.

## 3. Results

In total, 150 people (77.7%) returned the questionnaire and were included in the analysis (Figure 1). In the few cases where the responses were obtained by telephone interview or with support in the Rehabilitation Unit, the interview was performed within 1–2 weeks after referral. The median response time among e-mail responders was 5 days (*n* = 110, 14 missing), and IQR was 18 days. Among mail responders, the median response time was 33 days (*n* = 25, 12 missing) and IQR was 32 days.

Table 2 summarizes characteristics of the total population as well as data stratified by survey participation.

The majority of the study population were men (68.9%), with a mean age of 67.0 years. Approximately one-third lived alone (29.3%), while 34.6% had low educational attainment. Health literacy mean scores ranged from 2.67 (scale 5) to 3.70 (scale 6). The survey participants were more highly educated than nonparticipants (70.1% versus 45.7%), and though only close to significant, less often lived alone (26.0% versus 42.9%). Just below one-fifth (19.2%) of the survey participants did not attend the offered rehabilitation program. As may be expected in a group of people with a long-term condition, both physical and mental HRQoL summary scores were well below the general population mean of 50 points (40.0 and 46.2 points, respectively).

Table 3 shows the results of the unadjusted and adjusted logistic regression analyses on the four socio-demographic measures and each mean HLQ scale score in relation to rehabilitation participation.

None of the socio-demographic measures was significantly associated with the odds of participating in rehabilitation. Similarly, a one-unit increase in any of the nine mean HLQ scale scores was not significantly associated with the odds of participation in rehabilitation. There was a non-significant trend in eight of the nine scales that higher health literacy levels increased the odds of participation in rehabilitation.

Table 4 shows the unadjusted and adjusted linear regression analyses on the four socio-demographic measures and each mean HLQ scale score in relation to physical and mental HRQoL.

In the adjusted analyses, none of the four socio-demographic measures showed significant associations with either PCS or MCS. By contrast, a one-unit increase in mean HLQ was positively associated with higher PCS for scales 6 and 7, with β coefficients of 3.53 (0.88;6.18) and 2.79 (0.12;5.46), respectively. A one-unit increase in mean HLQ was also positively associated with higher mental MCS across five scales (scales 4, 6, 7, 8, and 9), with β coefficients of 6.61 (3.53;9.68), 4.63 (1.63;7.64), 7.10 (4.36;9.83), 4.83 (2.00;7.66), and 9.64 (6.09;13.18) respectively.

## 4. Discussion

In a population of people referred to a municipal cardiac rehabilitation program, we found no association between health literacy and participation in rehabilitation. We showed that several aspects of health literacy were positively associated with physical and mental components of HRQoL. Finally, we showed that age, gender, cohabitation, and educational attainment were not associated with participation in rehabilitation or with physical and mental components of HRQoL.

*Interpretations.* To a large extent, our population resembles other populations in cardiac rehabilitation, with a large proportion of men and a relatively high mean age [1]. Compared with a similar study in a general Danish population [41], the HLQ scores of our population are low for scales 5 through 9, while the mean score for scale 3 (actively managing my health) was somewhat high in comparison. In summary, the population generally has low “cognitive capacities”, while the level of “executive capacities” is more mixed (cf. Table 1).

Very few studies have been conducted on health literacy in relation to rehabilitation. In contrast to our findings, Dankner et al. show that some cognitive capacities related to health literacy may play a significant role in rehabilitation participation [22]. In eight of nine HLQ scales, our results trended towards positive associations between health literacy and participation. The non-significance may simply have been a consequence of the small sample size. However, since the majority of the population was referred by a clinician (general practitioner or physician at hospital), a possible association between health literacy and participation may also have been clouded by the tendency to simply do as prescribed. We do not have any data providing evidence of the subsequent adherence to the rehabilitation program or the resulting health behavior changes, but it is possible that participants with low health literacy adhere less from the services offered. Thus, a recent Danish randomized controlled trial testing a patient education intervention with several health-literacy-sensitive features in cardiac rehabilitation has also shown promising results regarding rehabilitation adherence [24]. To inform health-literacy-sensitive initiatives in cardiac rehabilitation, future research using longitudinal designs in larger populations should further explore the possible associations between health literacy and participation and adherence to cardiac rehabilitation.

Our results on the positive association between health literacy and physical HRQoL (PCS) most likely reflect the complex reality of many people with extensive physical health problems, where the demands placed on their health literacy may be multiple. These findings are consistent with several other studies in people with cardiac disease [13,19,42]. Strong associations were found in relation to actively engaging with healthcare providers and navigating the healthcare system (scales 6 and 7), that is, “executive capacities”, which may reflect the many players and services that are often involved in the care of people with physical health challenges. An earlier study in people with cardiovascular disease also found associations between the more “cognitive capacities” of understanding health information (scale 9) and PCS [13]. However, in our study, this association did not reach statistical significance. Thus, even though most cardiac rehabilitation programs focus on educating patients about disease and risk, the learning needs that people with low PCS present with may also be associated with their ability to act upon this knowledge and use the health system in an appropriate way [36,41].

In line with our results, literature on general populations confirms strong associations between health literacy and mental HRQoL (MCS) [9,43]. Studies based on populations with cardiac disease are less conclusive [13,19]. However, none of these studies includes very comprehensive health literacy measures. Our results thus add evidence to the nature of this probable association.

We found a very strong association between the “social capacity” of having social support (scale 4) and MCS. This was perhaps to be expected, as social networks and mental well-being are closely related [44]. However, the finding is important to consider in relation to health literacy interventions, as research has shown that relations and social networks are important in mitigating the consequences of low health literacy [45,46]. As with PCS, we also found strong associations between MCS and the “executive capacities” of actively engaging with healthcare providers and navigating the healthcare system (scales 6 and 7) and with the ability to find good health information (scale 8). Furthermore, MCS was associated with the “cognitive capacity” of understanding health information well enough to know what to do (scale 9). It is not possible to infer causality in these associations, but we conjecture that there may be a vicious cycle between the different aspects of low health literacy, where they mutually reinforce each other. If so, all three types of capacities should be targeted simultaneously if successful improvements are to be achieved [36].

Reviews of international literature have shown that socio-demographic characteristics are strong predictors of low participation in rehabilitation [3] but are not necessarily predictors of rehabilitation outcomes such as quality of life [47]. The evidence in this area is still insufficient [1]. Surprisingly, in our study we found that socio-demographic characteristics were not strongly associated with participation in rehabilitation or physical and mental components of HRQoL. To some extent, these results may reflect our relatively small homogenous sample and our single-site setting. In any case, these factors are difficult to act upon within the constraints of a rehabilitation service. Thus, other indicators of vulnerability may be more useful in this context.

Based on our results, actions to improve individual health literacy or respond to individual health literacy needs through organizational change or individualized service delivery may offer a promising opportunity to improve the impact of cardiac rehabilitation. Our data indicate that broad measures of health literacy provide a comprehensive picture of the challenges faced by people referred to cardiac rehabilitation. These measures could potentially inform the development of more targeted services such as individualized care plans. Attempts to use HLQ scales to inform intervention development have been reported in different settings [48,49,50]. Future research should focus on evaluating the use and effect of these and other tools and interventions in this field.

*Strengths and limitations.* To the best of our knowledge, this is the first study to explore multiple dimensions of health literacy among people referred to cardiac rehabilitation. The use of the nine-scale HLQ is a strength of the study as it provides comprehensive evidence of the specific challenges in this population.

Low health literacy probably increased the likelihood of nonresponse to our survey, since responding to questionnaires is in itself a health literacy challenge. We sought to counter this issue by offering support when filling out the questionnaire and reminding nonresponders to reply. Our high questionnaire response rate (77.7%) is an indication of some success in this regard. However, a statistically significant difference was identified regarding educational attainment in the group of responders compared to nonresponders. If education is interpreted as a proxy for health literacy, this may indicate that the problem may not have been fully eliminated. Other studies confirm the role of educational attainment in health awareness and beliefs [51,52].

The lack of statistical strength due to the low number of people included in the study (*n* = 150) and the even smaller number reporting on their nonparticipation in the rehabilitation program (*n* = 28) is likely to have affected our results regarding health literacy and participation in rehabilitation. It is also likely that a relatively large share of the nonresponders in the survey were also nonparticipants in parts or the entire rehabilitation program. However, we have no data to support this assumption.

All our data are cross-sectional and do not allow causal inference. Health literacy is a dynamic capacity, and we did not have the opportunity to adjust for the time interval between the distribution of the questionnaire and the response. Thus, in some cases, responders may have concluded parts of the rehabilitation program before responding to the questionnaire, potentially skewing their health literacy levels compared with quick responders. In the case of rehabilitation participation, this may have increased the health literacy difference between participants and nonparticipants.

## 5. Conclusions

Using a broad health literacy measure covering nine aspects of health literacy across “cognitive”, “executive”, and “social capacities”, we provide a comprehensive analysis of some the challenges faced by people referred to cardiac rehabilitation. Our results indicate that health literacy may not be associated to participation in cardiac rehabilitation programs, while it is associated to some aspects of physical and mental HRQoL. Responding to peoples with diverse health literacy needs could be a potential target of future interventions targeting participation and outcomes in cardiac rehabilitation. We encourage further research to investigate the role of health literacy in cardiac rehabilitation. Results can be used to develop and pilot interventions using local information on health literacy and health literacy responsiveness.

## Figures and Tables

**Figure 1 ijerph-17-00443-f001:**
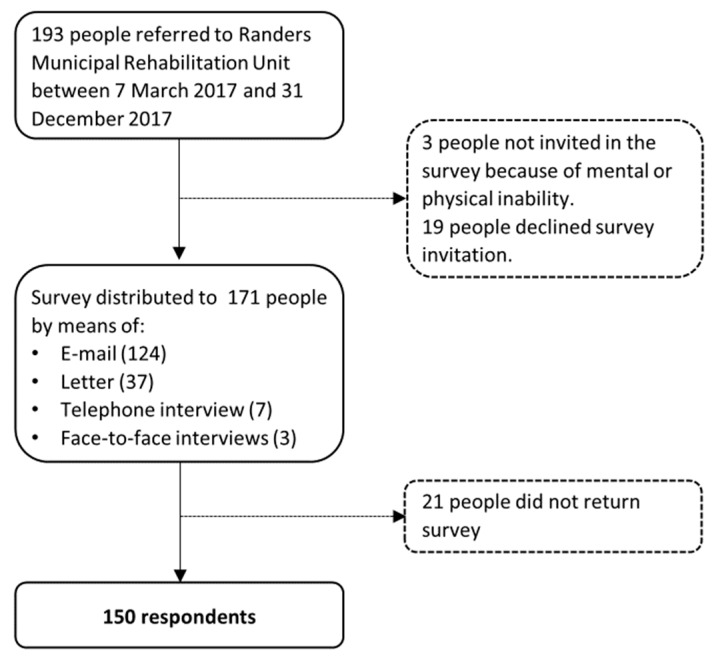
The recruitment of survey respondents for the Heart Skills Survey in Randers Municipal Rehabilitation Unit, 2017.

**Table 1 ijerph-17-00443-t001:** The nine HLQ scales characterized by capacity.

HLQ Scale	Category
2. Having sufficient information to manage my health (four questions)	Cognitive capacities
5. Appraisal of health information (five questions)	
9. Understand health information enough to know what to do (five questions)	
8. Ability to find good health information (five questions)	Executive capacities
3. Actively managing my health (five questions)	
6. Ability to actively engage with healthcare providers (five questions)	
7. Navigating the healthcare system (six questions)	
1. Feeling understood and supported by healthcare providers (four questions)	Social capacities
4. Social support for health (five questions)	

HLQ: Health Literacy Questionnaire.

**Table 2 ijerph-17-00443-t002:** Population characteristics by survey participation in the Heart Skills Survey in Randers Municipal Rehabilitation Unit (2017).

		Total Population (*n* = 193)		Survey Respondents (*n* = 150)		Survey Non-Respondents (*n* = 43)		*p*-Value
**Socio-Demographic Characteristics**								
Mean age (SD)		67.0	(12.1)	67.0	(11.8)	66.8	(13.4)	0.92
Gender	Male (%)	133	(68.9)	107	(71.3)	26	(60.5)	
	Female (%)	60	(31.1)	43	(28.7)	17	(39.5)	0.17
Country of origin	Danish origin	273	(96.1)	N/A		N/A		
	Non-Danish origin	7	(3.9)	N/A		N/A		
Cohabitation	Lives with someone (%)	128	(70.7)	108	(74.0)	20	(57.1)	
	Lives alone (%)	53	(29.3)	38	(26.0)	15	(42.9)	0.05
Educational attainment	Above 11 years (%)	117	(65.4)	101	(70.1)	16	(45.7)	
	11 years or below (%)	62	(34.6)	43	(29.9)	29	(54.3)	<0.05
**Health-Related Characteristics**								
Participation in rehabilitation	Participate (%)	N/A		118	(80.8)	N/A		
	Does not participate (%)	N/A		28	(19.2)	N/A		
Mean physical HRQoL (PCS) (SD)		N/A		40.0	(10.7)	N/A		
Mean mental HRQoL (MCS) (SD)		N/A		46.2	(11.1)	N/A		
Mean weighted Charlson Index (SD)		1.58	(1.4)	1.61	(1.4)	1.49	(1.2)	0.64
**Health Literacy Characteristics**								
2. Having sufficient information to manage my health								
Mean (SD)		N/A		3.00	(0.52)	N/A		
5. Appraisal of health information								
Mean (SD)		N/A		2.67	(0.50)	N/A		
9. Understand health information enough to know what to do								
Mean (SD)		N/A		3.61	(0.65)	N/A		
8. Ability to find good health information								
Mean (SD)		N/A		3.56	(0.70)	N/A		
3. Actively managing my health								
Mean (SD)		N/A		2.92	(0.48)	N/A		
6. Ability to actively engage with healthcare providers								
Mean (SD)		N/A		3.70	(0.66)	N/A		
7. Navigating the healthcare system								
Mean (SD)		N/A		3.38	(0.72)	N/A		
1. Feeling understood and supported by healthcare providers								
Mean (SD)		N/A		2.99	(0.57)	N/A		
4. Social support for health								
Mean (SD)		N/A		3.10	(0.52)	N/A		

SD: Standard deviation; significance level *p* > 0.05. HLQoL: Health-Related Quality of Life. PCS: physical component summary. MCS: mental component summary.

**Table 3 ijerph-17-00443-t003:** Associations between socio-demographic and health literacy measures and rehabilitation participation in the Heart Skills Survey in Randers Municipal Rehabilitation Unit (2017).

	Participation in Rehabilitation			
Socio-demographic characteristics	Crude		Adjusted *	
	OR	(95% CI)	OR	(95% CI)
Age (per year)	0.99	(0.95;1.02)	1.00	(0.96;1.04)
Gender (ref. women)	1.74	(0.74;4.11)	1.48	(0.57;3.82)
Cohabitation (ref. lives alone)	2.35	(0.97;5.69)	2.32	(0.90;6.01)
Educational attainment (ref. ≤ 11 years)	0.78	(0.30;2.01)	0.62	(0.23;1.69)
Health literacy characteristics	Crude		Adjusted **	
	OR	(95% CI)	OR	(95% CI)
2. Having sufficient information to manage my health	1.23	(0.53;2.83)	1.39	(0.55;3.53)
5. Appraisal of health information	1.50	(0.61;3.71)	1.49	(0.56;4.01)
9. Understand health information enough to know what to do	1.19	(0.61;2.33)	1.47	(0.69;3.12)
8. Ability to find good health information	1.29	(0.70;2.38)	1.57	(0.77;3.21)
3. Actively managing my health	1.67	(0.68;4.14)	1.72	(0.64;4.58)
6. Ability to actively engage with healthcare providers	1.33	(0.71;2.51)	1.36	(0.69;2.66)
7. Navigating the healthcare system	1.24	(0.66;2.33)	1.51	(0.74;3.08)
1. Feeling understood and supported by healthcare providers	1.89	(0.85;4.18)	1.74	(0.74;4.07)
4. Social support for health	0.90	(0.40;2.06)	0.97	(0.41;2.31)

OR: odds ratio; CI: confidence interval; significance level *p* > 0.05. * Adjusted for age, gender, country of origin, cohabitation, educational attainment, and comorbidity, excluding the independent variable in question. ** Adjusted for age, gender, country of origin, cohabitation, educational attainment, and comorbidity.

**Table 4 ijerph-17-00443-t004:** Associations between socio-demographic and health literacy measures and health-related quality of life.

	Physical Health Status (PCS)				Mental Health Status (MCS)			
**Socio-Demographic Characteristics**	**Crude**		**Adjusted ***		**Crude**		**Adjusted ***	
	**β **	**(95% CI)**	**β **	**(95% CI)**	**β **	**(95% CI)**	**β **	**(95% CI)**
Age (years)	−0.13	(−0.29;0.03)	−0.02	(−0.18;0.14)	0.10	(−0.06;0.27)	0.13	(−0.05;0.31)
Gender (ref women)	2.54	(−1.45;6.54)	2.78	(−1.34;6.90)	1.50	(−2.65;5.65)	1.23	(−3.31;5.78)
Cohabitation (ref single living)	3.99	(−0.09;8.08)	2.68	(−1.53;6.89)	1.93	(−2.38;6.24)	1.24	(−3.41;5.89)
Educational Attainment (ref ≤ 11 years)	2.47	(−0.48;6.43)	1.61	(−2.26;5.48)	−0.63	(−4.77;3.51)	−0.50	(−4.77;3.77)
**Health Literacy Characteristics**	**Crude**		**Adjusted ****		**Crude**		**Adjusted ****	
	**β **	**(95% CI)**	**β **	**(95% CI)**	**β **	**(95% CI)**	**β **	**(95% CI)**
2. Having sufficient information to manage my health	3.12	(−0.36;6.59)	2.56	(−0.93;6.05)	3.05	(−0.55;6.66)	2.87	(−0.96;6.71)
5. Appraisal of health information	1.93	(−1.63;5.49)	1.59	(−1.93;5.10)	1.82	(−1.91;5.54)	1.23	(−2.65;5.11)
9. Understand health information enough to know what to do	3.17	(0.34;6.00)	2.81	(−0.14;5.76)	6.06	(3.25;8.86)	6.61	(3.53;9.68)
8. Ability to find good health information	2.51	(−0.07;5.10)	1.59	(−1.22;4.41)	4.38	(1.76;6.99)	4.63	(1.63;7.64)
3. Actively managing my health	2.79	(−0.98;6.55)	3.04	(−0.63;6.71)	3.36	(−0.54;7.25)	2.72	(−0.33;6.77)
6. Ability to actively engage with healthcare providers	3.90	(1.25;6.55)	3.53	(0.88;6.18)	7.29	(4.75;9.84)	7.10	(4.36;9.83)
7. Navigating the healthcare system	3.34	(0.88;5.80)	2.79	(0.12;5.46)	4.93	(2.45;7.40)	4.83	(2.00;7.66)
1. Feeling understood and supported by healthcare providers	0.70	(−2.40;3.81)	0.42	(−2.66;3.50)	1.70	(−1.51;4.91)	0.96	(−2.42;4.35)
4. Social support for health	1.92	(−1.59;5.42)	1.13	(−2.43;4.70)	9.31	(6.00;12.62)	9.64	(6.09;13.18)

CI: confidence interval; significance level *p* < 0.05. * Adjusted for age, gender, ethnicity, cohabitation, educational attainment, and comorbidity, excluding the independent variable in question. ** Adjusted for age, gender, ethnicity, cohabitation, educational attainment, and comorbidity.

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
