# Peer review of "Health Literacy among People in Cardiac Rehabilitation: Associations with Participation and Health-Related Quality of Life in the Heart Skills Study in Denmark"

_ijerph, 2020, doi:10.3390/ijerph17020443_

Round 1

Reviewer 1 Report

Dear Authors and Dear Editor,

Thank you to giving me the opportunity to review this interesting paper.

Health literacy (HL) is a main topic in several domains of public health. To assess the association of HL and some health outcomes is relevant for healthcare personnel, as HL could be improved and modified through specific programs and strategies.

The paper reports the result of a survey among patients referred to cardiac rehabilitation service in Denmark.

The main results of the present paper are:

HL is not associated with participation in cardiac rehabilitation programs Some aspects of HL are associated with physical and mental HRQoL

A secondary result of the study is that common socio-demographics characteristics are neither associated to participation in rehabilitation programs nor to HRQoL

The study is well conducted and reported clearly. Few minor considerations should be discussed before publication process, in my opinion.

Suggestions:

The tile includes results about association of HL and HRQoL. No information about HL and participation in rehabilitation programs are reported. According to authors judgment, I suggest: “Health literacy and cardiac rehabilitation: association with participation and with health-related quality of life – cross-sectional results from the Heart Skills study in Denmark”. Abstract: “and more often lived alone”…..this sentence is based on data from Table 2 (Cohabitation). The level of significance here is 0.05 exactly. The findings are reasonable, but are reported in the abstract while are not discussed clearly in results/discussion section of the manuscript. Please, leave this information in abstract but state it in the text clearly, although significance is “weak” (in a numerical point of view). Methods: “to citizens with or in risk of a cardiac condition”. Please, clarify this sentence. Could be interpreted as source of bias? Is this solved through adjustment for comorbidity? Probably yes. But try to facilitate this concept for readers. Figure 1: check number. 193 people. 3 not invited and 17 declined. 171 surveys distributed (2 missing subjects). “we included data from the year of referral………..being recorded for that individual”. Please reformulate and improve fluency. Table 2: Participation in rehabilitation: 118 yes, 28 no = 146 (not 150….why?). Discussion: “in a population of patients (or subjects)”. Discussion: “However, a statistically significant ……..has been fully eliminated”. In my opinion, this concept is very relevant. The unique significant socio-demographic variable founded in this study is that low education is associated to low participation in the study (survey). As Authors stated, this could be interpreted as a consequence of poor HL itself. I should try to stress this concept underlying that a poor HL in study participation could have also consequences on health decisions of patients.

Try to add this concept. My suggestion is:

These data are in line with previous results that confirm how educational level could interfere with some aspects of cognitive, executive and social capacities of HL acting on participants decisions markedly, ranging from the participation to a clinical study itself [1] to some practical consequences on patients’ health management and beliefs [2,3]”

[1] Leiter A, Diefenbach MA, Doucette J, Oh WK, Galsky MD. Clinical trial awareness: Changes over time and sociodemographic disparities. Clin Trials. 2015 Jun;12(3):215-23. doi: 10.1177/1740774515571917.

[2] Sirico F, Miressi S, Castaldo C, Spera R, Montagnani S, Di Meglio F, Nurzynska D. Habits and beliefs related to food supplements: Results of a survey among Italian students of different education fields and levels. PLoS One. 2018 Jan 19;13(1):e0191424. doi: 10.1371/journal.pone.0191424. eCollection 2018.

[3] Janki Shankar, Eugene Ip, Ernest Khalema, Jennifer Couture, Shawn Tan, Rosslynn T. Zulla, Gavin Lam. Education as a Social Determinant of Health: Issues Facing Indigenous and Visible Minority Students in Postsecondary Education in Western Canada. Int J Environ Res Public Health. 2013 Sep; 10(9): 3908–3929. doi: 10.3390/ijerph10093908.

Conclusion: “Our results indicate that…….in cardiac rehabilitation”. Please reformulate. These are not direct conclusions of the paper. Conclusion are that HL is not associated to participation in cardiac rehabilitation programs, while it is associated to some aspects of physical and mental QoL”.

Regards

Reviewer 2 Report

This study seeks to investigate the link between L Health literacy (HL) and cardiovascular diseases. There is little doubt, as Authors state in the Introduction, that HL is a promising target of health equity interventions in non-communicable disease prevention. However, the sample chosen is not the best to evaluate the association between HL and prevention. In fact, it appears that patients have been referred to the centre by the treating physicians. In this situation, the individual level of HL has little influence, if any, on the decision to perform rehabilitation. In fact, even a person with low HL generally participates in a health activity prescribed by a doctor. However, we are not sure that he/she continues to do this activity unless he/she understands its usefulness.

Probably a longitudinal study could have shown a relationship between HL and adherence to the rehabilitation program. This could be a research commitment for the future, which the authors could indicate. At the moment, however, this is a conceptual limitation of the work that should be discussed.

For the above-reported reasons, it’s not surprising that Authors found no significant associations between HL and participation in cardiac rehabilitation.

The association between HL and physical and mental HRQoL is interesting and in line with the literature.

Reviewer 3 Report

This is an elegant study exploring an interesting and practically relevant issue. I have only one comment: although the Authors stated that their results are contradictory to those reported by Dankner et al. (ref. No. 42) with regard to the relation between health literacy-related cognitive capacities and participation in a rehabilitation programme, they have failed to provide possible explanations for this discrepancy.
